# Cost-Effectiveness and Cost-Utility of Palbociclib versus Ribociclib in Women with Stage IV Breast Cancer: A Real-World Data Evaluation

**DOI:** 10.3390/ijerph20010512

**Published:** 2022-12-28

**Authors:** Nour Hisham Al-Ziftawi, Mohammed Fasihul Alam, Shereen Elazzazy, Asrul Akmal Shafie, Anas Hamad, Mohamed Izham Mohamed Ibrahim

**Affiliations:** 1Clinical Pharmacy and Practice Department, College of Pharmacy, QU Health, Qatar University, Doha P.O. Box 2713, Qatar; 2Department of Public Health, College of Health Sciences, QU Health, Qatar University, Doha P.O. Box 2713, Qatar; 3Pharmacy Department, National Center for Cancer Care and Research, Hamad Medical Corporation, Doha P.O. Box 3050, Qatar; 4School of Pharmaceutical Sciences, Universiti Sains Malaysia, Gelugor 11800, Malaysia

**Keywords:** cyclin-dependent-kinase 4/6 inhibitors, advanced breast cancer, cost-effectiveness, cost-utility, cost-saving

## Abstract

Palbociclib and ribociclib are indicated in the first-line treatment of hormonal-receptor-positive HER-2 negative (HR+/HER-2 negative) advanced breast cancer. Despite their clinical benefit, they can increase healthcare expenditure. Yet, there are no comparative pharmacoeconomic evaluations for them in developing countries, the Middle East, or Gulf countries. This study compared the cost-effectiveness of palbociclib and ribociclib in Qatar. A 10-year within-cycle-corrected Markov’s model was developed using TreeAge Pro^®^ software. The model consisted of three main health states: progression-free (PFS), progressed-disease (PD), and death. Costs were obtained from the actual hospital settings, transition probabilities were calculated from individual-patient data, and utilities were summarized from the published literature. The incremental cost-effectiveness ratio (ICER) and the incremental cost-utility ratio (ICUR) were calculated and compared to three gross-domestic-products per capita. Deterministic and probabilistic sensitivity analyses were performed. Ribociclib dominated palbociclib in terms of costs, life-years gained, and quality-adjusted life-years gained. The conclusions remained robust in the different cases of the deterministic sensitivity analyses. Taking all combined uncertainties into account, the confidence in the base-case conclusion was approximately 60%. Therefore, in HR+/HER-2 negative stage IV breast cancer patients, the use of ribociclib is considered cost-saving compared to palbociclib.

## 1. Introduction

Breast cancer is one of the most common non-communicable diseases worldwide, ranking second amongst all cancers with a total estimated number of 627,000 deaths (6.2% of total cancer-related deaths and 15% in women’s cancer-related deaths) in 2018 [1,2]. Stage IV breast cancer, including advanced breast cancer (ABC) and metastatic breast cancer (MBC), is challenging due to being incurable with low survival rates [3]. The majority of stage IV patients are observed to be hormone receptor-positive (HR+)/human epidermal receptor 2-negative (HER2-) [4]. For these patients, there are several first-line treatments depending on the patient’s case. Generally, endocrine therapy is considered the mainstay first-line treatment for the majority of patients with HR+ advanced breast cancer. Chemotherapy can also be used as a first-line treatment for patients who have a life-threatening disease or who require early relief of symptoms due to significant visceral organ involvement with endocrine [5]. Nonetheless, cyclin-dependent kinase 4 and 6 enzyme inhibitors (CDK4/6 inhibitors) are a relatively new class of medications that were approved in first-line treatment alongside endocrine therapy for HR+/HER-2 negative stage IV breast cancer patients after proving their clinical superiority compared to endocrine monotherapy [5]. To date, there are three CDK4/6 inhibiting agents approved by the Food and Drug Administration (FDA): palbociclib, ribociclib, and abemaciclib [6]. CDK4/6 inhibitors are expensive, and they need frequent close monitoring due to their side effects, which include blood-related side effects (such as neutropenia, febrile neutropenia, thrombocytopenia, anemia, and leukopenia), heart-related side effects (such as affecting the QT interval and induced abnormalities in electrocardiography), gastric side effects (such as diarrhea and constipation), and generalized fatigue and neurological pain [7,8]. Therefore, although CDK4/6 inhibitors were proven to add clinical benefit as per the available randomized controlled trials and observational studies [9,10,11,12,13,14,15,16], they can increase healthcare expenditure and healthcare costs. Thus, their use should be guided by reliable pharmacoeconomic evidence.

As for the pharmacoeconomic evidence, on one hand, to date, there are a few existing comparative cost-effectiveness studies for the cost-effectiveness of palbociclib and ribociclib in stage IV HR+/HER2-negative breast cancer patients [17,18,19,20,21]. The settings and perspectives of these studies varied from different countries, including the Spanish National Health System perspective in Spain [17], the third-party payer perspective in the United States of America (USA) [18,19], the National Health Services and Personal Social Services perspective in the United Kingdom (UK) [20], and the private healthcare system perspective in Brazil [21]. All of them concluded that the treatment with ribociclib plus letrozole was cost-effective compared to palbociclib plus letrozole, except for one study that concluded that both palbociclib plus letrozole and ribociclib plus letrozole combinations are not cost-effective compared to letrozole monotherapy [19]. Of note, all of these evaluations included only a single combination with the treatment of interest, palbociclib plus letrozole versus ribociclib plus letrozole, without taking any of the other FDA-indicated combinations into consideration, such as fulvestrant or tamoxifen. In addition, these analyses were based on secondary data obtained from phase III clinical trials that were mainly MONALEESA-2 [9], PALOMA-1 [16], and PALOMA-2 [12]. On the other hand, a recent 3-year cost-minimization analysis from Russia using more real-world data concluded that palbociclib is a cost-saving option compared to ribociclib, assuming equal clinical benefit [22]. Currently, only one comparative cost-effectiveness analysis compared the three CDK4/6 inhibiting agents in combination with fulvestrant in the USA [23]. It concluded that abemaciclib was a more cost-effective option than ribociclib, but it was less cost-effective than palbociclib [23]. Therefore, it can be concluded that the conclusion regarding the cost-effectiveness of CDK4/6 inhibitors varies depending on the different settings and perspectives from which the pharmacoeconomic analysis is carried out.

Qatar is an Arab country with a diverse population, where more than 80% of the population is not Qatari and comes from different ethnicities [24]. In Qatar, breast cancer remains challenging, as it is the most common type of cancer, accounting for 31% of the total new cases of cancer in 2018 [25]. The healthcare system in Qatar is a nonprofit healthcare system in which it is the main payer for healthcare services to all citizens and residents [26]. Cancer care is mainly provided by the National Center for Cancer Care and Research (NCCCR), which is the premier hospital for managing cancer in the state of Qatar and one of the main hospitals under Hamad Medical Corporation [27]. For stage IV HR+/HER-2 negative breast cancer patients, CDK4/6 inhibitors are used in the treatment of these patients; however, only palbociclib and ribociclib are authorized in the formulary so far. Due to the nature of the healthcare system in Qatar, where the government pays 100% of the cancer care on behalf of patients, and due to the fact that CDK4/6 inhibitors can increase healthcare costs, it is important to have strong cost-effectiveness evidence to guide their optimal use. To date, there are no cost-effectiveness analyses comparing palbociclib and ribociclib conducted in Qatar, nor in countries with similar healthcare systems and economic situations to Qatar, such as the Gulf countries—the regional, intergovernmental, political, and economic countries union comprising Bahrain, Kuwait, Oman, Qatar, Saudi Arabia, and the United Arab Emirates, or the Middle East. The existing cost-effectiveness analyses conducted worldwide may be misleading when adopted to Qatar due to the different perspectives, populations, and economic profiles. Therefore, this study aimed to compare the cost-effectiveness of palbociclib and ribociclib with their approved FDA combinations in stage IV HR+/HER-2 negative breast cancer females in the state of Qatar, which can also serve as a guide for countries with similar healthcare and economic profiles to Qatar.

## 2. Materials and Methods

### 2.1. Overview

A 10-year within-cycle-corrected Markov decision analytical model was developed to estimate overall costs, effectiveness (represented in life years gained), and quality-adjusted life years gained for the targeted population. The model was carried out from the healthcare-payer perspective, Hamad Medical Corporation, NCCCR. The incremental cost-effectiveness ratio (ICER) was compared to a willingness-to-pay (WTP) threshold of fewer than three times the national annual gross domestic product (GDP) per capita, as per the World Health Organization (WHO) for cost-effective interventions [28]. As a result, a treatment regimen of an incremental cost of less than QAR 576,150 per QALY gained was considered to be cost-effective and very cost-effective if it was less than QAR 192,050 per QALY, based on the Qatari GDP/capita of USD 52,751 (USD 1 = QAR 3.65, 2020 financial year) [29]. All costs and outcomes were discounted by an annual discounting rate of 3.5%. 

The model was composed of three health states: ‘progression-free disease’, which was defined in this model as the length of time in months that a patient can live with breast cancer while receiving palbociclib or ribociclib but without dying from tumor progression or following adverse events of the treatment; ‘progressed disease’, which is the length of time a patient can live with breast cancer after developing any increase in tumor size or new development of lymphadenopathy or distant metastasis; and ‘death’, which is the absorbing state in this model. All patients were assumed to enter the model in the ‘progression-free disease’ state. The transition between the health states followed a unidirectional transition, where at the end of each cycle, a patient could stay in the same state, move to the next state, or move directly to death, with no back transition to the previous state; this is due to the disease nature, since stage IV breast cancer is not curable. The Markov cycle length was assumed to be one month since that is the normal evaluation for the event development as per clinical guidelines. A visualization of Markov’s model for this study is illustrated in Figure 1. The model was developed and analyzed using the TreeAge Pro 2020.2.1^®^ software.

### 2.2. Model Inputs

The model inputs included costs, transition probabilities between health states, and effectiveness parameters (life months and utilities). These were estimated based on individual patient data and based on the published literature to feed the model as follows:Costs

The total direct medical costs per patient per month for each health state in each treatment strategy were calculated based on individual patient data inputs. The unit costs were obtained from the department of accounting and finance in Hamad Medical Corporation based on the 2019/2020 financial year. All costs were calculated in the local currency, Qatar riyals (USD 1 = QAR 3.65). For each of the two comparators, palbociclib and ribociclib, all cost components were analyzed based on the following components: drug acquisition cost, combination drug acquisition cost, laboratory tests needed throughout the treatment period complete blood count (CBC), blood chemistry tests (comprehensive metabolic panel, liver function test, magnesium and phosphorus levels), endocrinology tests (25-hydroxyvitamin D, TSH receptor antibody, follicle-stimulating hormone, vitamin B12), tumor markers and catechol amines (thyroglobulin and carcinoembryonic antigen, CEA), coagulation tests, urine analysis tests, clinical radiology costs (X-ray, ultrasound, mammogram, magnetic resonance imaging (MRI), computed tomography (CT), positron emission tomography scan (PET scan), and the bone density DEXA scan), the required cardiac procedure for the CDK4/6 inhibitors costs (electrocardiogram (ECG) and echocardiogram), costs of the outpatient visits, and hospitalization costs.

Effectiveness-based transition probabilities

The monthly transition probabilities were calculated from the real evidence of individual patient data based on HR+/HER2- stage IV breast cancer patients who were taking palbociclib or ribociclib from January 2016 to January 2020 at the NCCCR in Qatar. Firstly, cumulative probabilities of each of the events of interest for the two comparators were calculated based on Equation (1), where *P* is the probability and *A* is the event of interest. Secondly, the cumulative probabilities were converted into a rate as per the below Equation (2), where *P* is the cumulative probability, *r* is the rate, and *t* is the time in years [30]. Lastly, the rate was converted back to a 1-month transition probability as per Equation (3) [30]. Therefore, three unique transition probabilities with two complementary ones for each arm were generated as follows: monthly transition probability from PFS to PD, monthly transition probability from PFS to death, their complementary probability of staying in the PFS, monthly transition probability from PD to death, and its complimentary monthly probability of staying PFS for both groups. All transition probabilities, utility values [31,32,33], and discounting rates [34] are summarized in Table 1.

Cumulative probability of event (*A*) for a patient cohort.
(1)$$P\;\left(A\right)=\frac{Total\;number\;of\;cohort\;with\;event\;\left(A\right)}{Total\;number\;of\;cohort}$$

Constant rate from probability.
(2)$$r=-\frac{\ln\left(1-P\right)}{t}$$

Fixed time probability.
(3)Time probability (A)=1−exp (−r t)

Utilities

Quality of life was used to investigate the impact of the quality of life (QoL) on every additional year gained by the treatment of palbociclib and ribociclib to generate quality-adjusted life years (QALYs). QoL values were obtained from the published literature [31,32,33]. QoL values for the ‘progression-free disease’ health state were summarized from the published literature from findings from the PALOMA-2 trial for the palbociclib group [31] and from the MONALEESA-3 trial for the ribociclib group [32]. For the ‘progressed disease’ health state, it was assumed that there was no difference in terms of the quality of life between palbociclib and ribociclib. This assumption was based on the fact that when a patient develops a progression, she is managed according to the same hospital guidelines depending on the progression type she had and regardless of the CDK4/6 inhibiting drug that she received before. Therefore, the same utility value of 0.45 was used as per a published literature systematic review [33]. All utility values and other model inputs are summarized in Table 1. 

### 2.3. Sensitivity Analysis

To address the impact of any uncertainties regarding the model inputs on the conclusion of the cost-effectiveness or cost-utility, a univariate deterministic sensitivity analysis was implemented with a single scenario for each variable assessment. The variables assessed through the deterministic sensitivity analysis were costs, transition probabilities from ‘progression-free disease’ to ‘progressed disease’, and quality of life for each health state. In order to ensure the robustness of the output, each of the variables of interest were varied separately while fixing the other model inputs, and ICER and ICUR were calculated accordingly. Then, a tornado analysis was generated to determine the variables that had the maximum effect on the cost-effectiveness conclusion. Deterministic sensitivity analysis inputs and source of the inputs’ boundaries are summarized in Table 2. 

Additionally, a probabilistic sensitivity analysis (PSA) was implemented using the Monte-Carlo simulation analysis based on 10,000 unique simulations. Therein, the parameters were input as probability distributions rather than just fixed values, and the different distributions of the different parameters were varied together to generate 10,000 different scenarios with possible outcomes. The detailed inputs of the Monte-Carlo analysis are summarized in Table 3. In addition, the incremental cost-effectiveness scatterplot (ICE), also known as incremental cost-effectiveness plane, was generated to illustrate the ratio of the simulations favoring ribociclib treatment versus palbociclib treatment and to present the overall uncertainty surrounding the base-case conclusion cost-effectiveness results. 

## 3. Results

As per Markov’s model, the 10-year cost of the palbociclib treatment strategy was QAR 372,663.3 per patient. In accordance, it yielded a gain of 71.62 life months (5.968 life years (LYs)) and, overall, gained quality-adjusted life years (QALYs) of 3.058 per patient (36.70 quality-adjusted life months), whereas, for the ribociclib treatment arm, the estimated 10-year cost was QAR 333,584.4 per patient. Similarly, the model produced a gain of 75.96 life months (6.330 gained life years) and 37.93 quality-adjusted life months per patient, with 3.160 QALYs per patient in the ribociclib treatment arm. The cost and effectiveness values were incremented to compare the two treatment options. When compared to palbociclib, ribociclib appeared to be more effective in terms of both LYs and QALYs gained, and it was less costly. The detailed values of the base-case overall cost and effectiveness are shown in Table 4.

Regarding the univariate deterministic sensitivity analysis, the base-case conclusion of having ribociclib as a cost-effective option remained robust against the variation of the cost of PFS for the palbociclib group. However, it was shown that palbociclib was not dominated as described previously at a cost of PFS that equaled QAR 9612.9 (17.33% reduction of the input cost), but ribociclib was still a cost-effective option compared to palbociclib in total (ICUR = QAR 5572.79/QALY gained per patient). Similarly, the conclusion of the cost-effectiveness and cost-utility of the two medications remained the same, keeping ribociclib dominant over palbociclib with ±20% adjustment in the total PFS cost of ribociclib, with costs ranging from QAR 302,750 to QAR 364,418 and 3.160 QALYs. As for the transition probabilities of PFS to PD in the palbociclib arm to vary within a range between 0.0459 and 0.05035, the conclusion of having ribociclib dominant over palbociclib remained robust. On the other hand, for the monthly probability of PFS to PD for the ribociclib group, ribociclib remained more cost-effective than palbociclib when the probability was varied according to the MONALEESA-2 trial; nonetheless, it was dominant only when the probability was more than or equal to 0.04537 (25% variation from the base-case probability). The uncertainty regarding the probability of PFS to PD in the ribociclib treatment arm was associated with an ICUR ranging from QAR 12,894.35/QALY to a dominance range suggesting that ribociclib is either a cost-effective option or a dominant option. Lastly, the utility associated with the PFS of palbociclib was varied according to the 95% CI range of the base case (0.7387–0.7627). The conclusion of the cost-effectiveness of ribociclib over palbociclib remained robust over that range of uncertainty, where ribociclib was dominant over palbociclib in all the uncertainty ranges. Similarly, the conclusion of the cost-effectiveness of ribociclib over palbociclib remained robust when the utility of PFS in the ribociclib group was varied by 18.5%, according to the standard deviation associated with the utility value as obtained from the literature. That is, ribociclib was dominant over palbociclib all over the uncertainty range; however, it did not dominate palbociclib at utility values less than 0.622, but it was cost-effective with an ICUR range from QAR 54,664.07/QALY to QAR 198,120.36/QALY. Lastly, the utility of progressed disease varied at ±20%, and the conclusion of the domination of ribociclib over palbociclib remained robust all over the range that ICER ranged from. A tornado diagram was generated to illustrate the effect of the individual factors’ uncertainty on the overall cost-effectiveness conclusion (Figure 2). For further clarification, the deterministic sensitivity analysis parameter, cost outputs, QALYs generated, and the overall cost-effectiveness decision are summarized in Table 5. 

Lastly, as for the probabilistic sensitivity analysis by the Monte-Carlo simulation, the mean (SD) lifetime cost of palbociclib was QAR 386,778.1 (196,998.1) with an average (SD) gained QALYs of 3.135 (0.8725). For the ribociclib treatment group, the mean (SD) lifetime cost according to the generated simulations was QAR 354,057.03 (152,369), with an average (SD) gained QALYs of 3.246 (1.0425). To graphically test the uncertainty around the base-case conclusion of preferring ribociclib over palbociclib, the ICE plot of ribociclib versus palbociclib was generated (Figure 3). According to the figure, 59.01% of the generated scenarios (presented as green dots) still favored ribociclib over palbociclib. Ribociclib was still dominant in 26.14%, and it was higher in cost but more effective (cost-effective) in 32.87% of the cases. Nonetheless, it was less costly but less effective than in 24.65% of the cases (cost-saving) and inferior to palbociclib in 15.16% of the cases.

## 4. Discussion

In this research, we aimed at identifying the long-term cost, long-term effectiveness, and incremental cost-effectiveness of two treatment strategies that are used in the first-line treatment of HR+/HER-2 negative stage IV breast cancer. Therefore, a 10-year Markov model was run to summarize the long-term cost (QARs), effectiveness (LYs), and utility (QALYs) for each of the two treatment strategies. Overall, in our base-case analysis, the treatment with ribociclib was dominant over palbociclib in terms of both ICER and ICUR. The finding of ribociclib being more cost-effective than palbociclib remained robust against all the one-way sensitivity analyses at the 3 GDP WTP threshold (cost-effectiveness threshold). For the 1 GDP WTP threshold (very cost-effectiveness threshold), only the uncertainty regarding one factor, the utility of PFS status in the ribociclib treatment arm, yielded an ICUR above 1 GDP (QAR 198,120.3/QALY), suggesting that ribociclib is not a very cost-effective option. However, compared to the recommended 3 GDP WTP threshold as per the WHO, ribociclib is still a cost-saving option compared to palbociclib, even with the uncertainty associated with that factor. Of note, our conclusion remained robust against the probabilistic sensitivity analysis that was associated with the combined uncertainties of all factors. Approximately 60% of the yielded hypothetical 10,000 scenarios in the Monte-Carlo simulation suggested that ribociclib is cost-effective compared to palbociclib. 

To date, and to our knowledge, there are four comparative pharmacoeconomic evaluations regarding the cost-effectiveness and the cost-utility of palbociclib and ribociclib. The findings of our study were consistent with three of them. That is, in a study conducted in Spain by Galve-Clavo et al. (2018) to evaluate the ICER and ICUR of ribociclib plus letrozole versus palbociclib plus letrozole, the former was associated with an ICER of EUR 1007.69 (QAR 4360.56) per every additional life year gained and an ICUR of EUR 1543.62 (QAR 6,679.69) per each QALY gained, at a threshold of EUR 30,000/QALY (129,818.59 QAR) [17]. Therefore, this study revealed that ribociclib was also cost-effective and cost-useful compared to palbociclib from the Spanish National Health System perspective [17]. In another study by Mistry R. et al. (2018) conducted in the USA also comparing ribociclib plus letrozole versus palbociclib plus letrozole versus letrozole monotherapy, ribociclib was dominant over palbociclib with a cost-saving of USD 43,037 and was still cost-effective compared to the letrozole monotherapy option [18]. That pharmacoeconomic analysis was conducted from the USA private third-party payer perspective at a WTP threshold of USD 198,000/QALY (QAR 720,918.06/QALY) [18]. Our findings were also consistent with one more pharmacoeconomic analyses by Suri G. et al. (2019) conducted in the UK, where ribociclib plus letrozole was also compared to palbociclib plus letrozole [20]. Their study reported that ribociclib plus letrozole was a cost-effective treatment strategy from the National Health Services (NHS) and Personal Social Services (PSS) perspective in the UK at a WTP threshold of EUR 30,000/QALY [20]. In only one cost-effectiveness study conducted in the USA, neither palbociclib nor ribociclib were cost-effective options, and the reason for this is that the ICER was calculated for each of the two comparators versus letrozole monotherapy [19]. There was no incremental cost-effectiveness ratio between the two CDK4/6 inhibitors, and therefore, none of them were cost-effective compared to letrozole monotherapy [19]. 

Although the previous pharmacoeconomic analyses were all of high quality, we could not rely on their findings to generate conclusions applicable to Qatar settings for multiple reasons. First, the generalizability of pharmacoeconomic analyses across countries is sometimes impaired due to the different sources of price weights among different countries [35] and due to different perspectives from which the pharmacoeconomic analyses take place [36]. Second, all used the published phase III clinical trials as the source for their simulated cohort, probabilities, effectiveness, and utility endpoints. Despite the success of the analysis, in the end, the use of these phase III trials themself is associated with some limitations since they were not designed to catch both clinical and economic endpoints. That is, in many of the pharmacoeconomic analyses based on RCTs, they tried to summarize the economic outcomes from the pre-collected primary clinical outcomes; thus, the sources of the economic data were not primary [35]. Both the MONALEESA and the PALOMA trials from which the four pharmacoeconomic analyses took their data were not predesigned to catch economic data. Therefore, we used a predesigned source of data to rely on for our economic analysis, the observational study conducted by our team earlier in 2021. Third, the four pharmacoeconomic analyses all compared the use of palbociclib versus ribociclib with only one of the indicated combinations, letrozole. This is because they used the same published phase III trial cohorts and interventions for their data. We sought a more thorough pharmacoeconomic analysis considering all the FDA-approved treatment combinations, especially since the COX regression conducted in phase I concluded no statistically significant differences in efficacy between the different treatment combinations. As a result, our analysis filled these gaps, providing a powerful pharmacoeconomic analysis that can be doubtlessly used for decision-makers in Qatar and other countries with similar health economic considerations.

Our study had several strengths. To begin, it was the first pharmacoeconomic evaluation focusing on the cost-effectiveness and the cost-utility of CDK4/6 inhibitors in Qatar, the Gulf region, and the Middle East and North Africa (MENA) region in general. Therefore, the findings of our current pharmacoeconomic analysis can be used locally and regionally in countries that have similar economic profiles and healthcare systems, with mild modifications to fit their context. Moreover, it is the first pharmacoeconomic analysis regarding these two treatments that was based on real-world evidence rather than just a simulation from clinical trials, avoiding all the disadvantages of modeling from clinical trials. Besides, it is the first pharmacoeconomic evaluation that compared palbociclib and ribociclib with all their FDA-indicated combinations; other analyses compared only the CDK4/6 inhibitors plus letrozole. In addition, it included pre/post-menopausal females in the cohort, unlike the other analyses that included only post-menopausal females as their cohort. Lastly, we performed an internal critical appraisal for our pharmacoeconomic study using the Quality of Health Economic Study (QHES) evaluation tool to assure the quality of the produced analysis; thus, we can assume that our results are assured of validity with minimal bias. However, our research had the main limitation that the base-case results were generated using observational real-world evidence, which itself has some limitations and potential uncertainties. However, we addressed this limitation by incorporating both deterministic and probabilistic sensitivity analyses that relied on phase III published RCTs. Our pharmacoeconomic conclusions remained robust against the uncertainties in both the deterministic and the probabilistic sensitivity analyses.

Based on our study findings, we have several recommendations for the current practice and future research. First, since ribociclib had a lower overall cost than palbociclib, although it had a higher acquisition cost in general, further evaluation of the consumption of the related resources needs to be conducted in the future with a larger sample size for both treatment arms. Second, all the published pharmacoeconomic agents included only palbociclib and ribociclib in their analyses. Abemaciclib is another CDK4/6 inhibitor that is under-addressed by pharmacoeconomic evaluations. Thus, more comparative pharmacoeconomic evaluations need to be conducted about this medication along with the other two medications in the same CDK4/6 inhibiting family.

## 5. Conclusions

Since their introduction to the market, the use of CDK4/6 inhibitors is increasing due to their proven clinical efficacy. Nonetheless, they can increase health expenditure due to their high acquisition cost and monitoring cost. Therefore, we underwent a thorough cost-effectiveness analysis using a well-designed Markov model. Ribociclib was more cost-effective than palbociclib at a 3 GDP threshold and at a 1 GDP threshold, suggesting that ribociclib should be a more favorable option over palbociclib to use in practice whenever applicable. This conclusion remained robust against the different single uncertainties as well as combined uncertainties. As a result, ribociclib was proven to be generally more cost-effective than palbociclib in the state of Qatar. This finding can be generalizable to countries with similar economic profiles, considerations, and cost drivers to Qatar. More similar pharmacoeconomic analyses that include the three CDK4/6 inhibitors (palbociclib, ribociclib, and abemaciclib) need to be conducted for more robust comparisons.

## Figures and Tables

**Figure 1 ijerph-20-00512-f001:**
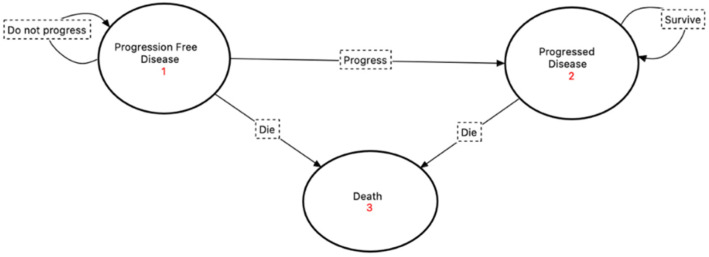
Visualization of the Markov model implemented in the study with the state diagram and their transitions’ pathways.

**Figure 2 ijerph-20-00512-f002:**
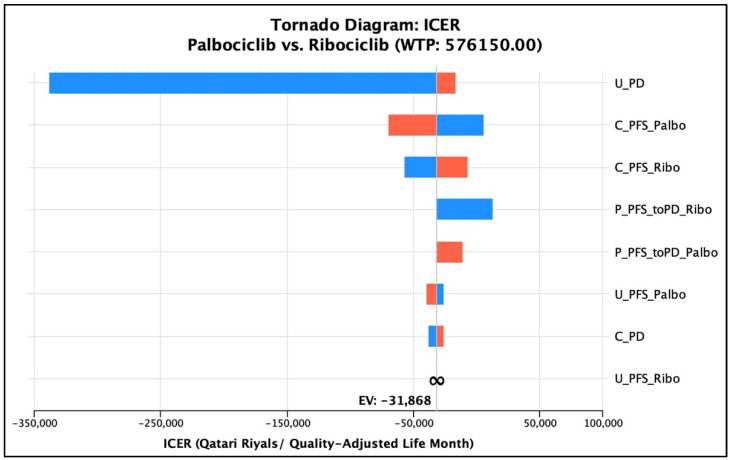
Tornado diagram of the univariate sensitivity analyses and their impact on ICER. The blue color represents the parameter, whereas the red color represents the ICER. U_PD: utility value of the ‘progressed disease’ health state; C_PFS_Palbo: cost of the ‘progression-free survival’ health state for the ‘palbociclib’ treatment group; C_PFS_Ribo: cost of the ‘progression-free survival’ health state for the ‘ribociclib’ treatment group; P_PFS_toPD_Ribo: transition probability from the ‘progression-free survival’ health state to the ‘progressed disease’ health state for the ‘ribociclib’ treatment group; P_PFS_toPD_Palbo: transition probability from the ‘progression-free survival’ health state to the ‘progressed disease’ health state for the ‘palbociclib’ treatment group; U_PFS_Palbo: utility value of the ‘progression-free survival’ health state for the ‘palbociclib’ treatment group; U_PFS_Ribo: utility value of the ‘progression-free survival’ health state for the ‘ribociclib’ treatment group; C_PD: cost of the ‘progressed disease’ health state.

**Figure 3 ijerph-20-00512-f003:**
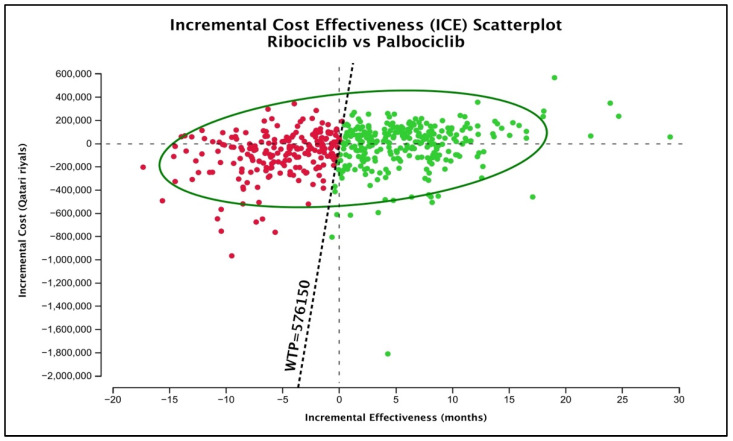
ICE plot of ribociclib versus palbociclib. Green dots represent the scenarios favoring ribociclib over palbociclib at a 3 GDP WTP of QAR 576,150.

**Table 1 ijerph-20-00512-t001:** Inputs of the Markov model.

Model Input	Value	Minimum	Maximum	SD	Source of Data
Costs/Months (QAR)					
PFS (palbociclib)	11,628.5	7477.4	14,316.4	-	RWD
PFS (ribociclib)	10,258.1	8926.4	11,054.2	-	RWD
PD	2942.6	1893.9	4118.3	-	RWD
Utility Values					
PFS (palbociclib)	0.7507	0.7387	0.7627	-	[31]
PFS (ribociclib)	0.710	-	-	0.185	[32]
PD	0.45	-	-	-	[33]
Monthly Transition Probabilities					
PFS to PD (palbociclib)	0.0459708	-	-	-	RWD
PFS to death (palbociclib)	0.0005916	-	-	-	RWD
PD to death (palbociclib)	0.0116347	-	-	-	RWD
PFS to PD (ribociclib)	0.0588690	-	-	-	RWD
PFS to death (ribociclib)	0.0029835	-	-	-	RWD
PD to death (ribociclib)	0.0063706	-	-	-	RWD
Discounting Rate					
	3.5%	1.5%	3.5%	-	[34]

PFS: progression-free survival; PD: progressed disease; RWD: real-world data.

**Table 2 ijerph-20-00512-t002:** Univariate deterministic sensitivity analysis (DSA) inputs.

Input Parameter	Base-Case Value	Sensitivity Analysis Boundaries		Source of Data
		Lower Boundary	Upper Boundary	
Cost of PFS state for palbociclib (QAR)	11,628.5	9302.8	13,954.2	±20% of base-case value
Cost of PFS state for ribociclib (QAR)	10,285.1	8228.1	12,342.1	±20% of base-case value
Cost of progressed disease state (QAR)	2942.6	2354.1	3531.1	±20% of base-case value
Monthly probability for PFS to PD in palbociclib	0.04597	0.04597	0.05036	[12,13]
Monthly probability for PFS to PD in ribociclib	0.05887	0.0261	0.05887	[9]
Utility of PFS state for palbociclib	0.7507	0.738	0.7627	[31]
Utility of PFS state for ribociclib	0.7	0.5705	0.8295	[32]
Utility of PD state	0.45	0.36	0.54	[33]

**Table 3 ijerph-20-00512-t003:** Probabilistic sensitivity analysis (PSA) inputs.

Input	Distribution	Point Estimate	Standard Deviation
Cost of PFS state for palbociclib (QAR)	Gamma	11,628.515	6838.95
Cost of PFS state for ribociclib (QAR)	Gamma	10,285.092	2127.73
Cost of PD state (QAR)	Gamma	2942.6	2224.34
Monthly probability for PFS to PD in palbociclib	Beta	0.04597	0.01364
Monthly probability for PFS to PD in ribociclib	Beta	0.05887	0.0260
Utility of PFS state for palbociclib	Beta	0.75	0.1290
Utility of PFS state for ribociclib	Beta	0.70	0.185
Utility of PD state	Beta	0.45	0.20

PFS: progression-free survival; PD: progressed disease; QAR: Qatari Riyal.

**Table 4 ijerph-20-00512-t004:** Base-case results for palbociclib and ribociclib treatment groups.

	Palbociclib	Ribociclib	Palbociclib Minus Ribociclib
Cost (QAR)			
Total cost	372,663.3	333,584.4	39,078.9
PFS cost	229,563.45	154,170.39	75,393.06
PD cost	143,099.89	179,414.02	−36,314.13
Effectiveness Outcomes			
Life years gained	5.968	6.330	−0.362
QALYs gained	3.058	3.160	−0.102
Cost-effectiveness			
ICER ICUR	- -	- -	Ribociclib dominated palbociclib Ribociclib dominated palbociclib

QAR: Qatari Riyal; PFS: progression-free survival; PD: progressed disease; ICER: incremental cost-effectiveness ratio; ICUR: incremental cost-utility ratio.

**Table 5 ijerph-20-00512-t005:** DSA outputs for palbociclib and ribociclib groups at each of the uncertainty parameters with the overall cost-effectiveness conclusions.

Uncertainty Parameter	Uncertainty Range	Palbociclib		Ribociclib		Cost-Effectiveness Conclusion
		Cost (QAR)	QALYs	Cost (QAR)	QALYs	
Base-case	-	372,663	3.058	333,584.4	3.160	Ribociclib dominates
Cost of PFS state for palbociclib (QAR)	+20% of base-case	418,576	3.058	333,584	3.160	Ribociclib dominates at PFS cost ≥ QAR 338,994
	−20% of base-case	326,750	3.058	333,584	3.160	ICUR = QAR 66,873/QALY
Cost of PFS state for ribociclib (QAR)	+20% of base-case	372,663	3.058	364,418	3.160	Ribociclib dominates
	−20% of base-case	372,663	3.058	302,750	3.160	Ribociclib dominates
Cost of progressed disease state (QAR)	+20% of base-case	401,283	3.058	369,467	3.160	Ribociclib dominates
	−20% of base-case	344,043	3.058	297,701	3.160	Ribociclib dominates
Monthly probability for PFS to PD in palbociclib	0.0459	372,663	3.058	333,584	3.160	Ribociclib dominates
	0.0503	355,998	2.986	333,584	3.160	Ribociclib dominates
Monthly probability for PFS to PD in ribociclib	0.0261	372,663	3.058	453,831	3.583	ICUR = QAR 154,723/QALY
	0.0588	372,663	3.058	333,584	3.160	Ribociclib dominates at a transition probability of ≥0.041
Utility of PFS state for palbociclib	0.738	372,663	3.037	333,584	3.160	Ribociclib dominates
	0.7627	372,663	3.078	333,584	3.160	Ribociclib dominates
Utility of PFS state for ribociclib	0.570	372,663	3.058	333,584	2.999	Ribociclib is cost-saving
	0.8295	372,663	3.058	333,584	3.322	Ribociclib dominates at a utility value ≥ 0.6223
Utility of PD state	0.36	372,663	2.694	333,584	2.703	Ribociclib dominates
	0.54	372,663	3.423	333,584	3.618	Ribociclib dominates

## Data Availability

All data generated or analyzed during this study are included in this article. Further enquiries can be directed to the corresponding author.

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
