# Peer review of "Cost-Effectiveness and Cost-Utility of Palbociclib versus Ribociclib in Women with Stage IV Breast Cancer: A Real-World Data Evaluation"

_ijerph, 2022, doi:10.3390/ijerph20010512_

Round 1
Reviewer 1 Report
The study is a detailed pharmacoenconomic evaluation of Palbociclib vs. Ribocyclib in Qatar. This issue is 'up-to-date', as the oncologists face the dilemma of which CDK4/6 inhibitor should be advised. Apparently, financial aspects should be taken into consideration, as the efficacy of all of these agents is comparable. The paper is interesting and well-written. However, I have some suggestions that hopefully improve this article.
1. Language: The paper is well written and understandable. I have no major concerns about the English style.
2. Major comments:
- There is a disproportion between introduction and discussion. In my opinion, the introduction should be shortened as some information is repeated in the discussion- mostly in the second paragraph of the introduction. Its sufficient to leave this data in the discussion, in the introduction only short information will be efficient.
- In the introduction, the actual place of CDK4/6 inhibitors in breast cancer treatment is not precisely specified. Provide information about visceral crisis / use of chemo vs. CDK4 / 6 inhibitors, etc.
- please add in the conclusion section some indications for clinicians that are the result of your study.
3. Minor comments
- in line 99 please explain what the Gulf countries are as it is not obvious for everybody
- in the tables, please avoid abbreviations or explain all of them below the tables.
4. Conclusions: In my opinion, the article deserves publication.
Author Response
Dear Reviewer,
Please find the revised ms and the responses attached.
Thanks.
Izham

Reviewer 2 Report
The economic evaluation is interesting and appears competently performed. However, I do have some concerns, mainly about the presentation.
· The title sets out the comparison of palbociclib to ribociclib. However, the wording in the text is not as clear in terms of the comparison as in the title. See for example the aim where it is not obvious to the reader what the intervention and the alternative is.
· A negative ICER cannot be interpreted and should not be calculated. I suggest that the authors either settles for presenting the incremental cost and benefits and concluding that ribociclib dominates Palbociclib, or uses the “net benefit” approach instead of the ICER.
· P2, lines 81-83: The authors state that the results in prior research varies over settings and perspectives. Then the authors should state the perspectives of the prior studies.
· I think the model could use a bit more explanation to help readers that are not experts on breast cancer. For example, why is there no path to cured/healthy, what does “progression free disease” actually mean, where does the costs occur (in 1, 2 or both 1 and 2), why is the costs so much lower for PD and why is there no difference between the drugs (this is mentioned later in the article), what is the effectiveness outcome? Etc.
· Table 2 includes information both relevant to the method and the results. If anything, it should be in the result section (currently method section), but be sure to explain in the method section how you vary the parameters in the sensitivity analyses (both deterministic and probabilistic) – this is currently not done.
· The text on page 6 (between tables 2 and 3) is hard to follow and should be rewritten for clarity. Also, the last sentence precedes the results.
· In the first paragraph in the result section, the authors calculate the average cost-effectiveness ration (cost/QALY) and compare this to the threshold. However, this is actually a comparison to a zero cost/zero effect alternative (i.e., immediate death), which is not a relevant alternative and therefore an inappropriate comparison. This should be excluded.
· Figure 2: I see no point to this graph, and it can be removed. In addition, it is inappropriately scaled in a way to exaggerate the difference between the drugs.
· P8, this is better presented in a table (Table 2) – but with “dominating” instead of negative ICER (if that indeed is the interpretation).
· I think the tornado diagram using net monetary benefit is a much better choice than ICER (you should consider excluding figure 3). But I don’t understand who the NMB is calculated. Given that the highest CIER I can see is 198k and with a threshold value of 576k, I cannot see how you could have gotten negative NMB. (NMB = (incremental benefit * threshold value) – incremental cost). Presumably, you have turned it around, but then it is unclear why at least one scenario give a positive NMB – but I guess it is possible the incremental costs could be high enough to give this result, but please clarify.
· As Figure 6 is based on Figure 5 (as I understand it), Figure 5 can be excluded. In Fig 6, I don’t understand the WTP line. Shouldn’t this be the threshold line, i.e., what society is willing to pay per QALY. If so, it should go through the coordinates for 576k and 12 months, but it doesn’t. So what is it and how it is related to conclusion about cost-effectiveness? Potentially related to this (unclear) is the colour coding in Fig 6. What does it mean? Is it scenarios where one drug is considered to be preferrable to the other? If so, it appears it is based on the wrong threshold.
Minor concerns
· Last sentence in the abstract states that “ribociclib is considered more cost-effective compared to palbociclib”. Since the comparison is explicit, the “more” here would refer to a comparison to another cost-effectiveness ratio. And since there is no other comparison being made, it is correct in this article to drop the “more” from the sentence. The same issue reappears in several places in the article (more/less) and should be changed accordingly.
· P. 2, lines 58-60: Incomplete sentence.
· P2, lines 62-63: “existing” not “existed”. Also at the end of the page.
· Regarding all graphs: there is no need to have decimals on the costs, it should be stated in the graphs what the effectiveness is (not just “effectiveness”), the variables names should be readable and meaningful (see 3 and 4).
· It’s base-case, not case-base.
· P12, lines 347-348: “we relied […] to rely”
Author Response

(The authors gave the same response as above.)

Round 2
Reviewer 2 Report
I think the authors have, overall, made a good revision of their manuscript. However, a few concerns remain (and a few new has come up) that I would encourage the authors to have a look at.
New comment
Clarify that the results presented in the first paragraph in the result section is “per patient”.
New comment
P. 8, line 236 is written “[…] it was still not cost-effective […]”. The “it” refers to palbociclib but should refer to ribociclib since ribociclib was the dominant option in the base-case.
New comment
Table 5: clarify what the comparison is, i.e. which treatment is dominant in the table and which treatment is “the first treatment” in the ICER calculation? Applies to the text surrounding table 5 as well. To be clearer: dominant refers to R compared to P, but the whole study is written in terms of P compared to R. I cannot see what the ICER in table 5 actually is P vs. R or R vs. P.
Original comment
A negative ICER cannot be interpreted and should not be calculated. I suggest that the
authors either settles for presenting the incremental cost and benefits and concluding that
ribociclib dominates Palbociclib, or uses the “net benefit” approach instead of the ICER.
Thank you so much. We really appreciate your comment. Therefore, we will still present
ICERs with “dominant” and “dominated” terms. You will find it adjusted all over the text
and highlighted in red.
Additional comment:
This is much approved in the revised version, but there are still a few negative ICERs presented in the result section that has no interpretation – if anything, the sensitivity would be better presented in terms of incremental costs and effects.
Original comment
P2, lines 81-83: The authors state that the results in prior research varies over settings and
perspectives. Then the authors should state the perspectives of the prior.
Thank you so much for your comment. We have already stated the detailed perspectives of
the existing studies while discussing them since it was more relevant to be mentioned there.
We have highlighted them in red for reference on lines: 342, 347, and 352.
Additional comment:
But you write explicitly on P2 “Therefore, it can be concluded[…]”, and if you conclude something you need to let the reader in on the information that have led to your conclusion. It’s a minor thing in the manuscript, but to discuss this 10 pages later is too late.
Original comment
I think the tornado diagram using net monetary benefit is a much better choice than ICER
(you should consider excluding figure 3). But I don’t understand who the NMB is calculated.
Given that the highest CIER I can see is 198k and with a threshold value of 576k, I cannot
see how you could have gotten negative NMB. (NMB = (incremental benefit * threshold
value) – incremental cost). Presumably, you have turned it around, but then it is unclear why
at least one scenario give a positive NMB – but I guess it is possible the incremental costs
could be high enough to give this result, but please clarify.
Thank you so much. We really appreciate your comment and understand where the concern
comes from. However, regarding the calculations of NMB, we did not do it as it was not one
of the objectives of this analysis. We just thought it would be good to have it represented in
the Tornado diagram to give more clarification which is an automatic option provided by
TreeAge Pro® using the same model inputs. We assume that one of the scenarios at least
gave a negative outcome depending on the variation cost, as you assumed. But we cannot
elaborate too much on this point because it might be outside the research scope. So we are
still willing to keep the tornado diagram represented in figure 3 previously (figure 2 now).
Alternatively, if you think the presence of NMB in this manuscript may be a source of
confusion, we can omit figure 3 (figure 4 previously) if you think it would be more
appropriate.
Additional comment:
I would then suggest that you drop the net monetary benefit diagram as the number does not align with the rest of the paper. For example, the baseline value is -745596 which does not align with the other figures presented. I wonder if there is a mix-up between the QALY over 1 month and over 1 year that leads to this? The same issue goes for the base-line value in the ICER tornado. (Also, see if you can simplify the notation for the different bars in the tornado diagram).
Original comment
As Figure 6 is based on Figure 5 (as I understand it), Figure 5 can be excluded. In Fig 6, I
don’t understand the WTP line. Shouldn’t this be the threshold line, i.e., what society is
willing to pay per QALY. If so, it should go through the coordinates for 576k and 12 months,
but it doesn’t. So what is it and how it is related to conclusion about cost-effectiveness?
Potentially related to this (unclear) is the colour coding in Fig 6. What does it mean? Is it
scenarios where one drug is considered to be preferable to the other? If so, it appears it is
based on the wrong threshold.
Thank you so much. We really appreciate your concern. As for figure 5, we agree with your
recommendation and have excluded it accordingly. Regarding figure 6 (now 4 after excluding
2 figures based on your valuable review), as we elaborated in the caption, green dots
represent ribociclib, and red dots represent palbociclib. The aim of this figure is to illustrate
the scenario distribution in terms of costs (Y-axis) and effectiveness (X-axis). The WTP
threshold line is to give more illustration, but it is not the major thing. As you can see, the
figure is divided into 4 quarters, and the scenarios are explained in terms of these 4 quarters.
This was also explained in the text but re-highlighted again in red for your consideration;
Lines 306 to 310.
Additional comment:
I think there might be some misunderstanding regarding the cost-effectiveness plan. My comment here is based on the standard interpretation of the CE-plan, if you have done something different, you need to explain this carefully as I think most readers will infer “my” interpretation. The purpose of the CE-plane is to show the sensitivity around the ICER, often by bootstrapping. Each dot represents a possible ICER within the sensitivity range. The dots do not represent one or the other of the alternatives but the comparison of the two alternatives (i.e., the incremental differences in costs and effects). The green dots represent those ICERs that are considered cost-effective, while the red dots are the not cost-effective. The threshold line is key in deciding what is considered cost-effective in the NE and SW quadrants. I realise now that my confusion the last time around was more about the effectiveness than the threshold line (the latter indeed looks correct). It says in the text that the effectiveness is QALY (not months as I read the first time around), but this needs to be clear in the graph. I’m surprised to see that the incremental QALY ranges from -20 to 30 given the average QALY is around 3. Are you sure that the effectiveness is not measured in months, as you do in the model? If so, the threshold should go through the point 576150:12 (see also my comment regarding the tornadoes).
I do like and agree with you revision regarding the proportion of ICERs that end up in each quadrant, but it would also be interesting to see how large proportion of ICERs are considered cost-effective.
Original comment
Last sentence in the abstract states that “ribociclib is considered more cost-effective
compared to palbociclib”. Since the comparison is explicit, the “more” here would refer to a
comparison to another cost-effectiveness ratio. And since there is no other comparison being
made, it is correct in this article to drop the “more” from the sentence. The same issue
reappears in several places in the article (more/less) and should be changed accordingly.
Thank you so much for your comment. Upon reviewing it, we found it valid and we have re-
visited the manuscript and dropped these words, and reworded it for clarity.
Additional comment:
This is now better but the change requires more editing in the text. In several places throughout the text it is now written “cost-effective than”, suggest changing to “cost-effective compared to”. Connected to this is that the result and discussion section needs to be revised to align with all the revisions that have been made.
Original comment
Regarding all graphs: there is no need to have decimals on the costs, it should be stated in the
graphs what the effectiveness is (not just “effectiveness”), the variables names should be
readable and meaningful (see 3 and 4).
Thank you so much for your comment. Regarding the decimals, unfortunately, we could not
change them since it is an original graph generated by TreeAge Pro® and this is how they
format it. Regarding the variables, we agree with your point. Still, since it is a graph and only
limited characters are allowed by variable, we have added an explanation downside to the
graphs regarding each of the variables’ abbreviations for clarity – highlighted in red in lines
276 – 285 and lines 290 to 299.
Additional comment:
The decimals are changed in the model configuration stored in Tree preferences -> Calculations -> Number formatting.
Author Response
Dear Editor,
Attached please find the responses to the comments and revised ms.
Thanks.
Izham
